# PARP Inhibitors Resistance: Mechanisms and Perspectives

**DOI:** 10.3390/cancers14061420

**Published:** 2022-03-10

**Authors:** Elena Giudice, Marica Gentile, Vanda Salutari, Caterina Ricci, Lucia Musacchio, Maria Vittoria Carbone, Viola Ghizzoni, Floriana Camarda, Francesca Tronconi, Camilla Nero, Francesca Ciccarone, Giovanni Scambia, Domenica Lorusso

**Affiliations:** 1Institute of Obstetrics and Gynecology, Università Cattolica del Sacro Cuore, Largo Agostino Gemelli 8, 00168 Rome, Italy; elenagiudice6@gmail.com (E.G.); viola.ghizzoni@gmail.com (V.G.); giovanni.scambia@policlinicogemelli.it (G.S.); 2Department of Biomedical Sciences and Human Oncology, University of Bari Aldo Moro, Piazza Giulio Cesare 11, 70124 Bari, Italy; marica.gentile@libero.it; 3Department of Woman, Child and Public Health, Fondazione Policlinico Universitario A. Gemelli IRCCS, Largo Agostino Gemelli 8, 00168 Rome, Italy; caterina.ricci@policlinicogemelli.it (C.R.); luciamusacchio89@gmail.com (L.M.); vittoriacarbone@hotmail.com (M.V.C.); camilla.nero@policlinicogemelli.it (C.N.); francesca.ciccarone@policlinicogemelli.it (F.C.); 4Medical Oncology, Università Cattolica del Sacro Cuore, Largo Agostino Gemelli 8, 00168 Rome, Italy; floriana.camarda@gmail.com; 5Medical Oncology, Università Politecnica delle Marche, Via Tronto 10/a, 60126 Ancona, Italy; fratronca@gmail.com

**Keywords:** ovarian cancer, polyADP-ribose polymerase (PARP) inhibitor, PARP inhibitor resistance, DNA damage repair, homologous recombination, BRCA, replication fork

## Abstract

**Simple Summary:**

This review aims to analyze the emerging issue regarding PARP inhibitor’s resistance in tumors and their consequence on disease prognosis and treatment. Besides, we evaluate possible strategies and new therapeutic approaches to overcome PARPis resistance.

**Abstract:**

PolyADP-ribose polymerase (PARP) inhibitors (PARPis) represent the first clinically approved drugs able to provoke “synthetic lethality” in patients with homologous recombination-deficient (HRD) tumors. Four PARPis have just received approval for the treatment of several types of cancer. Besides, another three additional PARPis underlying the same mechanism of action are currently under investigation. Despite the success of these targeted agents, the increasing use of PARPis in clinical practice for the treatment of different tumors raised the issue of PARPis resistance, and the consequent disease relapse and dismal prognosis for patients. Several mechanisms of resistance have been investigated, and ongoing studies are currently focusing on strategies to address this challenge and overcome PARPis resistance. This review aims to analyze the mechanisms underlying PARPis resistance known today and discuss potential therapeutic strategies to overcome these processes of resistance in the future.

## 1. Introduction

Genomic instability is one of the enabling characteristics of tumorigenesis. Reactive oxygen species, ultraviolet lights, ionizing radiation, endogenous and synthetic compounds represent recognized causes of DNA double-strand break (DSB) [1]. BRCA1 is an essential factor in the DSBs repair through homologous recombination (HR). The BRCA1, survival-dependency can be partially overcome by concomitant loss of p53 [2]. In addition to BRCA1, several other genes encode factors involved in HR, such as BRCA2, PALB2, and RAD51. Many tumors display the deficiency of one of these genes, presenting the so-called ‘BRCAness’ phenotype characterized by the deregulation of HR (HRD) [3]. This phenotype represents a well-defined biomarker for predicting PARP inhibitors (PARPis) activity in several cancer types, including breast, ovarian, pancreatic, and prostate cancer. Approximately 50% of high grade serous ovarian cancers (HGSOC) present HR deficiency (HRD). About 10–20% of breast tumors (mainly, triple negative breast cancers), metastatic prostate cancers, and pancreatic cancers harbor biallelic mutations in HR genes, making these tumors candidates for PARPis therapy [4,5,6,7].

The principal mechanism through which BRCA1/2 deficient tumors show their susceptibility to PARPis is represented by the “synthetic lethality” [8]. BRCA1 and RAD51C promoter methylation results in transcriptional silencing and commonly leads to HRD in HGSOC. In preclinical studies, BRCA1 or RAD51C methylation led to increased sensitivity to PARPis and platinum [9], although it is difficult to establish an association between methylation and clinical responses [10]. Given the variety of mechanisms that can result in HRD, it is challenging to identify a single, standardized test to detect HRD tumors regardless of the mechanisms involved. HRD cancers exhibit high genomic instability, characterized by deletions of large genomic segments, and genome-wide loss of heterozygosity (LOH), among other genomic aberrations [11]. Next-generation sequencing (NGS) techniques can identify multiple patterns of genomic change, including copy number variations, insertions/deletions and rearrangements which are characteristic of HRD.

At present, the myChoice CDx and FoundationOne CDx assays are the only NGS-based, prospectively validated, commercially available tests for HRD status assessment. The Myriad’s myChoice CDx generates a genomic instability score (GIS) from LOH’s combined measurement, including telomeric allelic imbalance (TAI), and large-scale transitions (LST). Meanwhile, the instability score derives from genomic signatures assessment, including microsatellite instability (MSI) and tumor mutational burden (TMB), in the FoundationOne CDx test [12,13].

HGSOC with a BRCA mutation or BRCA1/RAD51C methylation have high levels of genomic LOH [11]. HGSOC with high LOH (≥16%) are correlated with improved response to platinum and PARPis, regardless of BRCA status [14]. However, when homologous recombination repair (HRR) is restored or other PARPis resistance mechanisms develop, genomic scars in HRD cancer cells do not disappear, making them unsuitable biomarkers of PARPis sensitivity in later lines of treatment [15]. Based on these reasons, understanding the mechanisms leading to HRD and how they could change during the natural history of disease is important to predict outcomes with PARPis therapy.

## 2. PARP Is Action

Several PARPis have been studied. In clinical practice, four of these have been approved (olaparib, rucaparib, niraparib, and talazoparib) [16,17,18], whereas three are currently under evaluation (veliparib, pamiparib, and fluzoparib) [19].

Their common mechanism of action consists of preventing the repair of single strand binding protein (SSBs), which is capable of compromising this repair system with the consequent replication forks collapse [20]. Besides, the DNA damage induced by PARPis is bound by PARP1. This process of PARP1 trapping consequently produces DNA protein crosslinks, which in turn trigger the collapse of replication forks, thus resulting in DSBs accumulation during the S phase of the cell-cycle. Normally, these errors are repaired by the HR system, which is lacking in HR-deficient tumor cells, thus finally resulting in cell death [21] (Figure 1).

Murai et al. demonstrated a relationship between the PARP1 trapping activity of these agents and their cytotoxicity; the most effective PARP trapping activity belongs to talazoparib, which is approximately 100 folds more potent than niraparib. In turn, the latter is capable of PARP1 trapping better than olaparib and rucaparib. In preclinical models, a limited PARP1 trapping activity has been related to veliparib, which, in contrast, showed the ability to inhibit PARylation, and fails to elicit the same level of synthetic lethality [22]. Moreover, it has also been observed that PARP trapping activity is inversely proportional to the maximum-tolerated dose (MTD) of PARPis; thus, more potent PARP trappers often have to be administered at lower doses [23]. Conversely, no relationship between PARP trapping activity and the most common adverse effects of PARPis has been observed [24].

## 3. Mechanisms of Resistance to PARPis

Despite the success of PARPis, the increasing use of these drugs in clinical practice raised the issue of PARPis resistance, and it remains a therapeutic challenge.

Understanding the mechanisms of resistance is of outmost importance to oppose and overcome resistance and improve therapeutic outcomes (Figure 2).

### 3.1. Restoration of HR Activity

In HR-deficient cells, the most common acquired PARPis resistance mechanism is represented by the restoration of HR capacity, which may occur through different processes [25].

### 3.2. Reversion Mutations

BRCA reversion mutations can restore protein function and lead to PARPis and platinum-based chemotherapy resistance [26].

Restoration of BRCA1/2 function occurs by genetic events that delete the frameshift with the consequent restoring of the open reading frame (ORF), leading to the expression of a functional nearly full-length protein. Besides, it can be caused by the genetic reversion of the inherited mutation, which also restores a full-length wild-type protein. In vitro, the exposure of BRCA1 and BRCA2 mutated cancer cells to cisplatin or PARPis favors the development of secondary genetic changes on the mutated allele which restore a functional protein and confer platinum and PARPis resistance [26,27]. This mechanism of resistance is clinically relevant for BRCA-mutated cancer patients treated with platinum therapy: up to 46% of platinum-resistant BRCA mutated HGSOC exhibits tumor-specific secondary mutations that restore the ORF of either BRCA1 or BRCA2. Moreover, multiple reversion events in BRCA1/2 genes have been reported as a mechanism of platinum-resistance in a study of whole-genome characterization of chemo-resistant ovarian cancer [27]. Moreover, the somatically-reverted BRCA1/2 alleles can be detected by analyzing the circulating free DNA (cfDNA) derived from patients undergoing PARPis or platinum therapy [28].

### 3.3. DNA Polymerase θ (POLQ) (Required for MMEJ) Could Be a Driver of Resistance

It has been demonstrated that reversion mutations restoring the native reading frame of the genes can occur in BRCA1-, BRCA2-, PALB2-, RAD51C-, or RAD51D-mutant cancers. In HR genes, two different types of reversion were identified: true reversions and second-site reversions. The latter were intragenic deletions joined by short regions (1–6 bp) of DNA sequence microhomology or accompanied by an insertion [29]. This microhomology-associated DNA sequence “scar” suggested that the predominant cause of reversion may be the DNA-repair processes involving regions of microhomology able to repair DSBs. Microhomology-mediated end joining (MMEJ) or single-strand annealing represent examples of these processes. DNA polymerase θ (Polθ, also known as POLQ) is a protein known to be involved in one of the DSB repair pathways called the “error-prone MMEJ pathway.” It is encoded by POLQ, and its expression has been demonstrated to be inversely correlated with HR activity in epithelial ovarian cancers. The loss of Polθ in HR-proficient cells leads to the upregulation of HR activity and RAD51 nucleofilament assembly. On the contrary, Polθ knockdown in HR-deficient cells is linked to enhanced cell death. These results are in line with the evidence of embryonic lethality observed in mice presenting the genetic inactivation of an HR gene (Fancd2) and Polθ. Besides, RAD51-mediated recombination is hampered by Polθ through its ability to bind RAD51. In this context, Polθ may represent a new druggable target for ovarian cancer treatment due to its synthetic lethal relationship with the HR pathway [30].

### 3.4. Restoration of BRCA1 via Other Mechanisms

Preclinical evidence showed a possible impact of ionizing radiation on the restoration of BRCA1 function; the use of ionizing radiation seems to stimulate the truncated but hypomorphic BRCA1 splice isoforms lacking exon 11 to form RAD51 and BRCA1 foci, thus indicating partial restoration of HR [31].

Another condition that has been linked to this restoration is carrying the BRCA1C61G mutation. It disrupts the N-terminal RING domain, conferring minimal residual activity in DNA damage responses and consequently inducing PARPis resistance [32].

Furthermore, the restoration of BRCA1 activity is also possible through the action of HSP90, which interacts with and stabilizes the C-terminal truncated BRCA1 protein [33]. This stabilization allows a partial function of the new stabilized protein and promotes RAD51 loading into DNA, thereby conferring PARPis resistance.

Cancer cell lines and tumors harboring mutations in exon 11 of BRCA1 express a BRCA1-Δ11q splice variant lacking the majority of exon 11. The introduction of frameshift mutations to exon 11 resulted in nonsense-mediated mRNA decay of full-length, but not the BRCA1-Δ11q isoform. CRISPR/Cas9 gene overexpression experiments revealed that the BRCA1-Δ11q protein may promote partial PARPis and cisplatin resistance relative to full-length BRCA1, both in vitro and in vivo [31]. Furthermore, spliceosome inhibitors reduced BRCA1-Δ11q levels and sensitized cells carrying exon 11 mutations to PARPis treatment. Wang et al. provided evidence that cancer cells employ a strategy to remove deleterious germline BRCA1 mutations through alternative mRNA splicing, giving rise to isoforms that retain residual activity and contribute to therapeutic resistance [34].

### 3.5. BRCA1—Independent Restoration of HR

Several other mutations may compromise regulation of DNA end-resection via loss of 53BP1, MAD2L2/Rev7 or the shieldin complex and enable HRR in the absence of BRCA1.

BRCA1 mutated cells result in less susceptibility to PARPis when 53BP1 is lacking; this phenomenon is explained by the 53BP1 capacity of restoring HR through the promotion of ATM activity which produces ssDNA targetable by HR [35]. Furthermore, loss of Lig4 can prevent the joining of DSBs into chromosome rearrangements, whereas it does not have the capacity to rescue HR in BRCA1 mutated cells [36]. Dev et al. demonstrated that the “shieldin” complex (SHLD1/2) consists of two proteins, C20orf196 and FAM35A. The latter presents a C-terminal OB-fold region that binds single-stranded DNA (ssDNA). This complex represents the downstream effector of 53BP1/RIF1/MAD2L2, which promotes end-joining DSB, and antagonizing the BRCA2/RAD51 is able to counteract HR [37]. The shieldin complex also interacts at the DNA damage sites, with the Christ–Siemens–Touraine complex (CST/Polα/πριμασε), which engages ssDNA in order to fill in the resected DSBs. This fill-in reaction, together with the CST/shieldin persistence at the DNA damage sites, is responsible of the blockade of HRD, with the consequent lethal mis-repair [38].

### 3.6. Stabilization of Replication Forks

BRCA1 and BRCA2 genes are involved in the fork protection of stalled replication. In the case of BRCA1-2 deficient tumor cells, the nucleases MRE1164 and MUS8165 are able to attack the stalled replication forks, with consequent fork collapse and chromosomal aberrations. When the PARPis resistance develop, a mechanism for inhibiting DNA replication fork degradation has been observed, caused by some nucleases able to stabilize the replication fork. In particular, in BRCA2-and BRCA1 deficient cells, EZH2 and PTIP activity respectively are downregulated at the fork, reducing the recruitment of nucleases and resulting in fork protection. PARPis in fact are known to induce fork degradation of an unprotected replication fork, and, as a consequence, increased stabilization of replication forks confers PARPis resistance [39].

### 3.7. Increased Drug Efflux

A murine model revealed another mechanism of PARPis resistance in BRCA1-deficient breast tumors. An overexpression of drug-efflux transporter genes (Abcb1a and Abcb1b) encoding for MDR1/P-gp and Abcg2 has been observed: increased drug-efflux enhanced the rate at which compounds, such as PARPis, are removed from cells and, as a consequence, their efficacy. Some evidence suggests that PARPis resistance may be related to the increased expression of drug-efflux transporter genes. This phenomenon is supposed to be specifically mediated by the Abc1a/b genes. A study showed that expression of Abcb1a/b was increased by 2- to 85-fold in olaparib-resistant breast cancers [40]. Abc1a/b expression was shown to be correlated with resistance to olaparib and rucaparib treatment in ovarian cancer cell lines, and cotreatment with the P-glycoprotein (P-gp) efflux pump inhibitor tariquidar resensitized the tumors to PARPis [41,42]. The resistance was also reversed following treatment with verapamil or elacridar, two common Abcb1a/b inhibitors [43]. However, Abcb1a/b overexpression was not shown to induce resistance to treatment with veliparib or AZD2461, an olaparib analog, indicating that this is unlikely to be the sole mechanism of PARPis resistance [43].

Despite strong preclinical evidence, no clinical trials have yet been designed targeting P-gp pump efflux for patients experiencing PARPis resistance [44]. In the near future, clinical trials in this setting are needed to confirm the relationship between increased expression of P-gp efflux pumps and PARPis resistance.

### 3.8. Inhibition of PARP Trapping and PAR Glycohydrolase (PARG) Mutations Can Lead to PARP Resistance

As described above, the PARPis ability of PARP trapping represents the meaning cause of cell death by the accumulation of unrepaired SSBs and impaired progression of replication forks [22]. Recent studies have demonstrated a functional link between the inhibition of PARP trapping activity and the onset of PARPis resistance [45]. Petit et al. showed a common mutation of PARP-1 (R591C) in PARPis-resistant tumor samples, which is linked to a diminished PARP1 trapping activity on DNA, thus resulting in PARPis resistance [31]. Besides, the PAR glycohydrolase (PARG) enzyme is also involved in PARP1 trapping activity, preventing polyADP-ribose (PAR) accumulation through the reversion of PARylation. Consequently, the loss of PARG leads to the accumulation of PAR in PARPis-treated cells, with the rescue of PARP1-dependent DNA damage signaling, thus reducing the activity of PARP1 trapping on DNA, finally resulting in PARPis resistance [45].

### 3.9. Alterations in Cell Cycle Control

Cyclin-dependent kinase 12 (CDK12) and WEE1 are cell-cycle regulators involved in PARPis resistance due to their ability to restore HR [46]. DNA repair proteins are downregulated by CDK12 knockdown, conferring sensitivity to PARPis, thereby inducing a phenocopied “BRCAness”. Besides, another mechanism that promotes PARPis sensitivity is mediated by the inhibition of WEE1. The abrogation of G2 cell-cycle checkpoint causes the premature entering of HR deficient cells into mitosis, with their unrepaired DNA damage and the consequent accumulation of DNA DSBs.

### 3.10. Dysregulated Signaling Pathways

Several molecular signaling pathways regulating cell division are involved in PARPis resistance. The proto-oncogene mesenchymal–epithelial transition tyrosine kinase reduces the binding capacity of PARPis, through its ability to activate the PARP1 enzymatic activity by phosphorylating PARP1 [47]. Furthermore, a significant upregulation of the PI3K/AKT pathway has been observed following the PARPis use, thus resulting in the promotion of cell growth and proliferation. Finally, the PARPis resistance is also linked to the upregulation of the ATM/ATR pathway. It represents an essential checkpoint of the DNA damage response process due to its capacity of recruiting DNA repair complexes through the phosphorylation of histone H2A. This phenomenon leads to the HR restoration [48], and thus, the inhibition of this pathway may represent a future strategy to overcome PARPis resistance.

## 4. Perspectives

Several studies are ongoing with the attempt to develop therapeutic strategies that elude the emergence of acquired resistance.

Potential strategies to fight PARPis resistance may include combination therapies aimed at amplifying the anti-tumor effects of PARPis by targeting the acquired vulnerabilities of PARPis resistant cancers through the suppression of the mutator phenotype.

### 4.1. Suppression of Alternative HR Pathways

It has been demonstrated that the recruitment of ssDNA by PALB2 is dependent on the RNF168 activity in BRCA1 mutated cells [49]. This phenomenon, together with the increased end-resection caused by the loss of the 53BP1–RIF1–REV7–shieldin axis, is responsible for the HR restoration in BRCA1/53BP1 double deficient cells [50]. Besides, the type of BRCA1 mutation determines the extent of HR restoration caused by the loss of 53BP1, because this reactivation is linked to the overall efficiency of RAD51 loading.

Therefore, the loss of RNF168, which mediates the PALB2 recruitment, is able to compromise HR and confer PARPis sensitivity in BRCA1 heterozygous cells and in BRCA1/53BP1 double deficient cells [51]. Targeting of RNF168 may represent a therapeutic option in order to improve PARPis efficacy, inhibiting BRCA1 independent PALB2/BRCA2 recruitment.

### 4.2. Suppression of Mutator Phenotype

The genomic instability deriving from the error-prone MMEJ pathway is suppressed by the POLQ inhibition. Novobiocin (NVB) is an antibiotic which is known to inhibit the ATPase activity of POLQ. The depletion of POLQ is linked to DSBs end resection, consequent accumulation of ssDNA, and finally leads to cell death in HR deficient tumors. In a PDX model, it has been demonstrated that NVB in combination with BRCA1-and-53BP1 loss of function reduces tumor growth [52]. Clinical trials are needed to understand the role of POLQ inhibition. POLQ depletion may be used as a novel anticancer strategy potentially superior to PARPis, because it may be effective in tumors with acquired resistance to PARPis, such as HR-deficient cancers of a ‘mutator phenotype’, and also in preventing or attenuating the onset of PARPis resistance in HR deficient naïve cells. In conclusion, POLQ inhibition may represent a promising strategy, combined with PARPis, or as an alternative to their use.

### 4.3. Indirect Inhibition of HR

Nowadays, there is no literature evidence of direct inhibitors able to catalyze HR. The inhibition of HR restoration following the treatment with PARPis may be mediated by some oncoproteins interfering with gene expression, nuclear localization and/or the recruitment of HR proteins, resulting in the indirect inhibition of HR. These oncoproteins represent promising actionable targets currently under study in combination with PARPis to overcome PARPis resistance.

For example, therapies targeting EGFR, IGF1R, VEGF or the PI3K/AKT pathway have been reported to impair HR [53].

Combining the VEGF antagonist bevacizumab with olaparib or niraparib improved the median progression free survival (PFS) in newly diagnosed and recurrent ovarian cancer patients, in both HRD and HR-proficient cohorts [54]. The NRG-GY004 demonstrated that the combination of cediranib and olaparib did not improve PFS compared to standard chemotherapy in women with platinum-sensitive ovarian cancer; however, post-hoc analysis suggested better activity in BRCA patients [55]. Moreover, the results from phase IIb, single-arm CONCERTO trial demonstrated antitumor activity of the cediranib-olaparib combination in heavily pretreated, non-gBRCAm patients, with recurrent platinum-resistant ovarian cancer [56].

Ongoing trials are addressing the combination of an oral anti-angiogenic agent, cediranib, with olaparib both in the platinum-sensitive (ICON9, NCT02345265) and platinum-resistant (NRG-GY005, NCT02889900, NCT02345265) setting. Additionally, an ongoing study is evaluating the addition of cediranib to olaparib in patients who progressed after initial response to therapy with olaparib alone (NCT02681237) [57].

Several preclinical studies have demonstrated synergistic anti-tumor activity with the combination of PARPis and PI3K/AKT pathway inhibitors in both BRCA deficient and proficient cancer models [58]. A phase I trial combining the AKT inhibitor capivasertib with olaparib revealed durable responses in patients with advanced-stage solid tumors, irrespective of BRCA1/2 status [59]. Another study showed a potential clinical activity with manageable toxicity with ceralasertib, an AKT inhibitor, in combination with olaparib in BRCA-mutated PARPi-resistant HGSOC, with a duration of benefit that exceeded the duration achieved on prior PARPi monotherapy [60,61].

Besides, the combination of PI3K inhibitors and PARPis has also been tested in clinical studies; in particular, the alpha specific PI3K inhibitor alpelisib (BYL719) has demonstrated to be safe in a multicenter, open-label, phase 1b trial [62] and a phase III trial is currently ongoing to investigate the efficacy of this combination in women with platinum resistant or refractory HGSOC, with no germline BRCA mutation detected (NCT04729387) [63].

However, this combination therapy does not directly inhibit HR; rather, it seems to impair cell-cycle progression. These data suggest an additive effect rather than a synergistic action of this combination therapy.

### 4.4. Immunotherapy in HRD Deficient Cancers

An encouraging strategy to fight PARPis resistance is represented by the combination of PARPis with immune checkpoint inhibitors. The rationale of this combination relies on the evidence that an increased mutational burden in HRD cancers may result in increased availability of tumor-specific neoantigens with consequent enhanced immunotherapy efficacy [64]. In fact, it has been observed that the expression of repetitive RNAs may activate immune signaling. In BRCA ½ deficient cells, the presence of these RNAs arises from the disruptions of the chromatin boundaries, which in turn is caused by the genomic rearrangements. PARPis induces both PD-L1 expression and cGAS–STING signaling, leading to increased CD8+ T cell infiltration and activation [65]. However, whether PARPi-mediated activation of cGAS–STING signaling is dependent on the BRCA mutation status of the tumor is still a matter of debate. Results from various studies indicate that PARPis enhance the anti-tumour effects of anti-PD-1 antibodies in breast and ovarian cancer mice models [66]. Ongoing clinical trials are evaluating the clinical impact of this combination therapy [67]. Results from Phase II clinical trials show that antibodies targeting PD-L1 in combination with PARPis exhibit a good response against germline BRCA1/2-mutated breast cancer and ovarian cancers [67,68].

Besides, this combination is currently under evaluation in several trials in newly diagnosed and recurrent (either platinum sensitive or resistant) ovarian cancer patients.

ENGOT-ov46/AGO/DUO-O is a phase III randomised, double-blind, multi-centre study evaluating the efficacy and safety of durvalumab in combination with standard of care platinum-based chemotherapy and bevacizumab followed by maintenance durvalumab and bevacizumab or durvalumab, bevacizumab and olaparib in patients with newly diagnosed advanced ovarian cancer [69]. Besides, the ENGOT-ov43/MK-7339/KEYLYNK-001 study is a randomised phase III, double-blind trial, aiming at investigating the role of immunotherapy (pembrolizumab) given in combination with chemotherapy and after the completion of the first-line treatment as maintenance therapy, given in combination with olaparib/placebo in patients diagnosed with BRCA non-mutated advanced epithelial ovarian cancer [70]. Another randomized, double-blind, phase III study, the ENGOT-OV44/FIRST study, is evaluating platinum-based therapy ± dostarlimab followed by niraparib ± dostarlimab maintenance as first-line treatment of advanced ovarian cancer [71]. ATHENA (GOG-3020/ENGOT-ov45) is a randomized phase III trial evaluating maintenance therapy after first-line treatment with rucaparib as monotherapy (ATHENA-MONO) vs. rucaparib in combination with nivolumab (ATHENA-COMBO) [72].

The role of PARPis in combination with immunotherapy in the platinum-resistance setting is under investigation in different trials (NItCHE-MITO 33 [73], MOONSTONE/GOG-3032 [74], NCT02485990 [75], even with the addition of an anti-angiogenic agent (BOLD [76] and OPAL trial [77], NCT04739800 [78], NCT02484404 [79]). The TOPACIO/KEYNOTE-162 study showed an ORR of 18% (90% CI, 11–29%), with a disease control rate of 65% (90% CI, 54–75%) with the combination of pembrolizumab-niraparib in platinum resistant, heavily pretreated OC patients [80].

Concerning the platinum-sensitive recurrence, the ANITA trial is a phase III randomized, double-blinded trial evaluating the efficacy of platinum-based chemotherapy with or without atezolizumab followed by niraparib maintenance +/− atezolizumab [81]. Furthermore, GINECO-EN203b/ENGOT-EN8 (ROCSAN trial) is a multi-centre randomized phase II/III trial, evaluating dostarlimab in combination with niraparib versus niraparib alone compared to chemotherapy in the treatment of endometrial/ovarian carcinosarcoma (recurrent or progressing) after at least one line of platinum-based chemotherapy [82].

### 4.5. Abrogation of Cell-Cycle Checkpoint Signalling

As described above, the ATM/ATR pathway represents one of the most involved checkpoints of the DNA damage response.

Combination therapy with PARPis and ATR inhibitors are under investigation as potential methods of overcoming resistance to PARPis in BRCA1 deficient tumors with restored HR function or fork protection [83].

The chromatin-remodeling enzyme ARID1A interacts with ATR in order to regulate the DNA damage checkpoint. Cells lacking ARID1A are consequently unable to activate this cell-cycle checkpoint, thereby acquiring susceptibility to PARPis treatment [84]. Besides, the first phase of HR is dependent on the ATM kinase activity, and its inhibition is linked to the susceptibility to PARPis in BRCA1-53BP1 and/or BRCA1-REV7 deficient cells [85].

Preclinical studies support the existing hypothesis that targeting other cancer-promoting pathway components (such as WEE1, MEK/MAPK, and PI3K) may enhance PARPis efficacy; in particular, while still preserving anti-tumor activity [86]. The phase II randomized, non-comparative EFFORT trial reported the efficacy of adavosertib (WEE1 inhibitor) in PARP-resistant ovarian cancer with an objective response rate (ORR) of 29% in the combination adavosertib plus olaparib arm and 23% with adavosertib alone [87]. The ongoing randomized phase I/II SOLAR (NCT03162627) trial combines olaparib with the MEK/MAPK/ERK inhibitor selumetinib in solid tumors. The rationale of this combination relies on the MEK inhibitors’ capacity to alter the apoptotic balance, induce HR deficiency, and decrease DNA damage checkpoint activity in RAS-mutant cells. These two agents synergistically act to increase DNA damage and apoptosis in response to PARPis [88].

The clinical applicability of these combinations, in the light of their toxicity profiles (Table 1).

### 4.6. Targeting Acquired Vulnerabilities

It has been demonstrated that despite the detrimental activity of PARG inactivation to PARPis efficacy, the loss of functional mutations involved in PARPis increases sensitivity to ionizing irradiation [45]. Furthermore, the loss of 53BP1–RIF1–REV7–shieldin or CST end-protection complexes and PARP1 loss may enhance the ionizing radiation effect [89]. As a consequence, radiotherapy might be a valid option for patients with BRCA deficient tumors with acquired resistance to PARPis. Besides, the combination of PARPis and radiotherapy may be of particular interest in the oligometastatic progression setting. In fact, the patients progressing on PARPis treatment with evidence of oligometastatic disease may benefit from local stereotassic radiotherapy, and prolong PARPis treatment beyond progression while achieving continuous benefit [90].

## 5. Conclusions

Several PARPis received FDA approval both in the frontline and second-line maintenance settings in ovarian cancer and in the treatment of other solid tumors such as pancreatic, prostate and breast cancer, thus enhancing the number of patients potentially benefitting from this strategy. However, despite the established clinical benefits across several solid tumor types, the increasing use of PARPis in clinical practice switch on the light on an emerging clinical issue which is represented by the PARPis resistance. The HR restoration, the DNA replication fork stabilization, the BRCA reversion mutations, the drug efflux increase, the dissociation of PARP1 and PARG, and the epigenetic molecular modifications are only some of the established mechanisms contributing to PARPis resistance. Preclinical models have offered insight into these mechanisms; still, analyses of paired tumor samples in early-phase clinical trials have suggested that multiple mechanisms may be inherent to progression on PARPis treatment. Ongoing clinical trials are evaluating the rationale of combination strategies to evade PARPis resistance. Promising combinations are represented by PARPis + anti-angiogenic agents, PARPis + immunotherapy, PARPis + PI3K/AKT pathway inhibitors, PARPis + Ras/Raf/MEK/MAPK pathway inhibitors, PARPis + ATR/WEE1 inhibitors, and PARPis + epigenetic modifiers. Although preliminary analyses suggested favorable response and clinical benefits, the short durability of response and the challenging toxicity profiles of the combined treatments are relevant clinical issues; in fact, it has to be noted that a combination treatment often involves more severe toxicity profiles. Further exploration of meaningful molecular markers of response is needed to delineate which patient will benefit most from each combination therapy, thus prompting the necessity to re-characterize from a molecular point of view the tumors progressing on PARPis treatment.

## Figures and Tables

**Figure 1 cancers-14-01420-f001:**
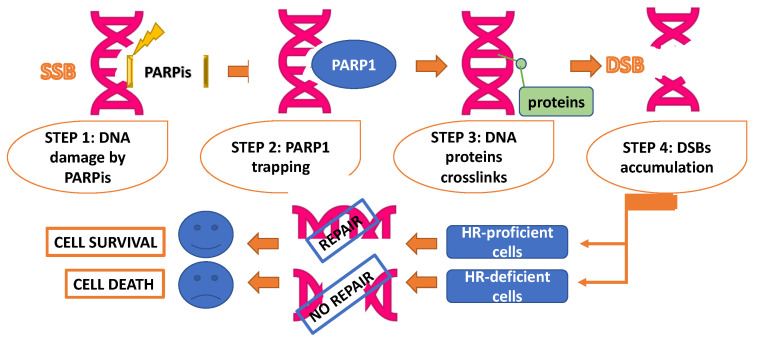
Mechanism of action of polyADP-ribose polymerase (PARP) inhibitors. Step 1: DNA damage caused by PARP inhibitors, with consequent creation of single strand DNA break (SSB); Step 2: detection of SSB by PARP1; Step 3: production of DNA protein crosslinks; Step 4: collapse of replication forks and double strand DNA breaks (DSBs) accumulation. While in homologous recombination (HR)-proficient cells these errors are restored by HR system, in HR-deficient tumor cells this process finally results in cell death.

**Figure 2 cancers-14-01420-f002:**
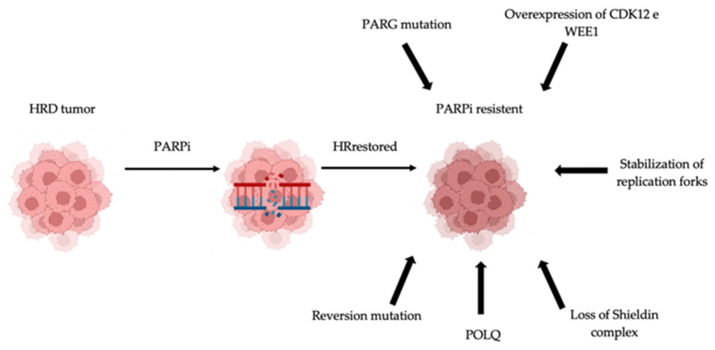
Mechanism of resistance to polyADP-ribose polymerase (PARP) inhibitors. Several resistance mechanisms have been identified: reversion mutation, DNA polymerase θ (POLQ), loss of shieldin complex, stabilization of replication fork, increased drug efflux, overexpression of cyclin-dependent kinase 12 (CDK12) and WEE1.

**Table 1 cancers-14-01420-t001:** Perspectives for eluding resistance to PARP inhibitors. Potential strategies to enhance the efficacy of poly(ADP-ribose) polymerase (PARP) inhibitors in treatment-resistant tumors.

Perspectives for Eluding PARPI Resistance	
Suppression of Alternative HR Pathways	
Suppression of mutator phenotype	Loss of RNF168
	POLQ inhibition by novobiocin kills HR deficient tumors in vitro/in vivo
Indirect inhibition of HR	CONCERTO trial: cediranib + olaparib
	AKT inhibitor: capivasertib + olaparib
	ceralasertib + olaparib
Immunotherapy in HRD deficient cancers	durvalumab + olaparib + bevacizumab (AGO-DUO) (BOLD)
	durvalumab + olaparib + cediranib
	pembrolizumab + olaparib (ENGOT ov43)
	pembrolizumab + niraparib (TOPACIO)
	atezolizumab + niraparib (ANITA)
	nivolumab + rucaparib (ATHENA combo)
	dostarlimab + niraparib (MITO33) (MOONSTONE) (ROCSAN)
	dostarlimab + niraparib + bevacizumab (OPAL) tremelimumab + olaparib
	tremelimumab + durvalumab + olaparib
Abrogation of cell-cycle checkpoint signalling	EFFORT trial: adavosertib (WEE1 inhibitor) + olaparib
	SOLAR trial: selumetinib (MEK inhibitor) + olaparib
Targeting acquired vulnerabilities	Ioniziting radiation

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
