# Peer review of "PARP Inhibitors Resistance: Mechanisms and Perspectives"

_cancers, 2022, doi:10.3390/cancers14061420_

Round 1

Reviewer 1 Report

This review is well written and summarized the PARP inhibitors usage nowadays as well as the perspective studies on overcoming the drug-resistant treatment. The paper is worthy to be published. However, some of the details need to be taken care of. Line 23, poli need to be poly. Line 107, the underlines in Figure 1 need to be removed.

Author Response

Reviewer comment: This review is well written and summarized the PARP inhibitors usage nowadays as well as the perspective studies on overcoming the drug-resistant treatment. The paper is worthy to be published. However, some of the details need to be taken care of. Line 23, poli need to be poly. Line 107, the underlines in Figure 1 need to be removed.

Authors’ Response: Thank you for your interesting comments and for your observations.

  • Line 23: “poli” is changed with “poly” as you suggest
  • Line 107: we removed the underlines in Figure 1 as you suggest

Reviewer 2 Report

In this manuscript by Giudice et al., the authors review the mechanisms and perspectives of PARP inhibitors resistance. It is a well-written, complete, and up-to-date revision of the mechanisms underlying resistance to PARP inhibitors. This referee considers the topic of the manuscript is of high interest to translational scientists and oncologists since it provides the perspectives being explored at clinical trials to overcome resistance to PARP inhibitors, which constitutes a major issue in the clinic.

This referee believes that some modifications should be done to the manuscript to improve its clarity to the reader.

Major Concerns:

  • This Reviewer strongly encourages the authors to include a figure describing the mode of action of PARP inhibitors. In section 2 (PARPis action) this reviewer believes that including a model of the mechanism of action of PARP inhibitors would help the reader interpret the manuscript.
  • Table 1 summarizes the potential strategies to enhance the efficacy of PARP inhibitors and overcome resistance. Please note that, at least in the pdf. manuscript provided to this referee, the columns and rows of this table are not arranged adequately. Because of this, it is not possible to clearly understand on which mechanisms each of the proposed perspectives is based.

Minor Concerns

Line 134: The full-length words for POLQ and MMEJ abbreviations have not been described before. Please clarify the meaning of POLQ or MMEJ or avoid using the abbreviations in the title.

Lin 225: The full-length word for PARG abbreviation has not been described before. Please clarify the meaning of PARG or avoid using the abbreviation in the title.

Line 248: The abbreviation for mesenchymal–epithelial transition (MET) is not required, since it is no longer mentioned in the manuscript.

Author Response

Reviewer comment: In this manuscript by Giudice et al., the authors review the mechanisms and perspectives of PARP inhibitors resistance. It is a well-written, complete, and up-to-date revision of the mechanisms underlying resistance to PARP inhibitors. This referee considers the topic of the manuscript is of high interest to translational scientists and oncologists since it provides the perspectives being explored at clinical trials to overcome resistance to PARP inhibitors, which constitutes a major issue in the clinic.

This referee believes that some modifications should be done to the manuscript to improve its clarity to the reader.

 Authors’ Response: Thank you for your interesting comments and for your observations.

Major Concerns:

  • This Reviewer strongly encourages the authors to include a figure describing the mode of action of PARP inhibitors. In section 2 (PARPis action) this reviewer believes that including a model of the mechanism of action of PARP inhibitors would help the reader interpret the manuscript.
  • Authors’ Response: Thank you for your suggestion. We added Figure 1 in order to help the reader to understand the mechanism of action of PARP inhibitors.
  • Table 1 summarizes the potential strategies to enhance the efficacy of PARP inhibitors and overcome resistance. Please note that, at least in the pdf. manuscript provided to this referee, the columns and rows of this table are not arranged adequately. Because of this, it is not possible to clearly understand on which mechanisms each of the proposed perspectives is based.
  • Authors’ Response: Thank you for your observation. Columns and rows of this table are now arranged adequately as you suggest.

Minor Concerns

Line 134: The full-length words for POLQ and MMEJ abbreviations have not been described before. Please clarify the meaning of POLQ or MMEJ or avoid using the abbreviations in the title.

Authors’ Response: Thank you for your suggestion, we included the full-length words for POLQ and we delete MMEJ abbreviation in the title.

Lin 225: The full-length word for PARG abbreviation has not been described before. Please clarify the meaning of PARG or avoid using the abbreviation in the title.

Authors’ Response: Thank you for your observation, we included the full-length words for POLQ.

Line 248: The abbreviation for mesenchymal–epithelial transition (MET) is not required, since it is no longer mentioned in the manuscript.

Authors’ Response: Thank you for your observation, we removed the abbreviation MET as you suggest.

Reviewer 3 Report

PoliADP-ribose polymerase (PARP) inhibitors (PARPis) are able to induce synthetic lethality in tumors that lack of homologous recombination (HR) activity. In this review, the authors make a summary including the mechanism of PARPis leading to cell death as a cancer therapy for HR deficient tumor, and the mechanisms of the development of PARPis resistance (reversion mutations, PolQ, other BRCA-dependent and independent restoration of HR etc.). Additionally, the authors also summarize the prospective strategies to treat PARPis resistance, which covers a lot of ongoing clinical trials that combine alternative HR pathway inhibitor, indirect HR inhibitors, immunotherapy, cell-cycle checkpoint inhibitors with PARPis, respectively. I think this manuscript in general is well organized and written. The information provided would be of great interests to the audiences of Cancers journal. Thus, I recommend it for publishing by this journal.

Specific comments:

In the manuscript, the authors mentioned PARPis have been used in breast cancer, ovarian cancer, pancreatic cancer, prostate cancer. Among the ongoing trials, most of them are for breast cancer and ovarian cancer. I was curious is there any other specific strategies for prostate and pancreatic cancers, or they are mostly the same?

Minor:

  1. line 110: Figure 1 legend: CDK12e

  1. Figure 1 remove red curly lines under the text

  1. line 120: nearly full-length

Author Response

Reviewer comment: PoliADP-ribose polymerase (PARP) inhibitors (PARPis) are able to induce synthetic lethality in tumors that lack of homologous recombination (HR) activity. In this review, the authors make a summary including the mechanism of PARPis leading to cell death as a cancer therapy for HR deficient tumor, and the mechanisms of the development of PARPis resistance (reversion mutations, PolQ, other BRCA-dependent and independent restoration of HR etc.). Additionally, the authors also summarize the prospective strategies to treat PARPis resistance, which covers a lot of ongoing clinical trials that combine alternative HR pathway inhibitor, indirect HR inhibitors, immunotherapy, cell-cycle checkpoint inhibitors with PARPis, respectively. I think this manuscript in general is well organized and written. The information provided would be of great interests to the audiences of Cancers journal. Thus, I recommend it for publishing by this journal.

Specific comments:

In the manuscript, the authors mentioned PARPis have been used in breast cancer, ovarian cancer, pancreatic cancer, prostate cancer. Among the ongoing trials, most of them are for breast cancer and ovarian cancer. I was curious is there any other specific strategies for prostate and pancreatic cancers, or they are mostly the same?

Authors’ Response: Thank you for your interesting comments. We expect that the mechanism of PARPi’s resistance are common among solid tumors and deeply inherent in the mode of actions of the drugs. As such, also the strategies to overcome resistance are expected to be similar. The literature data however, mostly refer to ovarian and breast cancer because of the long-term and extensive clinical use of the drugs in these diseases which raised the clinical issue.

Minor:

  1. line 110: Figure 1 legend: CDK12e
  • Authors’ Response: Thank you for your observation, we included the full-length words for CDK12 as you suggest
  1. Figure 1 remove red curly lines under the text
  • Authors’ Response: Thank you for your suggestion, we changed the red curly lines in the text as you indicate in the Figure (now Figure 2 and no more 1).

  1. line 120: nearly full-length
  • Authors’ Response: Thank you for your observation, we separated the two words as you indicated.